# Development of Variable Side Mold for Free-Form Concrete Panel Production

**Jongyoung Youn, Jiyeong Yun, Jihye Kim and Donghoon Lee \***

Department of Architectural Engineering, Hanbat National University, Daejeon 34158, Korea;
97colin@naver.com (J.Y.); 9736jy@naver.com (J.Y.); 6312904@naver.com (J.K.)
* Correspondence: donghoon@hanbat.ac.kr; Tel.: +82-42-821-1635

**Abstract:** With the increase of free-form architecture, many studies have been conducted for producing free-form Concrete Panels (FCP), but there are still areas that are lacking in terms of the technological aspect. In particular, as free-form panels are produced by hand, the precision of the shapes is low and cost and time are high. FCP production equipment was developed to resolve this. In this study, the variable side mold for FCP production used in side control equipment among FCP production equipment was developed. Variable side mold is equipment that satisfies five requirements to configure the form of FCPs. The variable side mold is made with steel plates so that it can withstand the side pressure of concrete. As a result, the material has a uniform thickness throughout and is molded to the desired shape. Therefore, in order to verify this, the panel was manufactured as a variable side mold to compare the side form with the designed form through 3D scanning and quality inspection to check for errors. As a result, there was a 0.276 mm mean difference for both ends of the panel and the central part, and it was therefore verified through *t*-test that errors occurred within the allowed margin of 95% confidence level.

**Keywords:** free-form concrete panels (FCP); variable side mold; FCP production

## 1. Introduction

With the increase in the amount of free-form architecture, there are studies being conducted on how to realize free-form architecture [1]. Free-form architecture has massive external parts, and it is difficult to produce this at a single time [2]. Currently, it is being produced through paneling work that separates it into smaller shapes [3]. Free-form panels require configuration to different forms depending on the parts that become panels, and it must be produced whilst considering structural safety, constructability, economic feasibility, etc. A material that satisfies these features and is superior to other free-form panels is concrete [4]. Free-form panels that use concrete are defined as free-form Concrete Panel (FCP). The side of the FCP has different shapes and curves depending on the separated parts, and in order to manifest this, a free-form mold must be custom-made. However, because free-form molds are made to the FCP form, they cannot be reused after they have been placed; therefore, they end up as construction waste. In addition, significant construction time is needed from placing to construction following customized mold production.

In fact, about 80,000 FCPs were used to build the disc-shaped architecture of the National Museum of Qatar [5]. There were different forms and sizes for the used FCP, so different molds were required, and thus 3000 molds were produced [6]. A total of 539 disc-shaped FCPs were used to create the shape of the National Museum of Qatar, and this required a long period of time [7]. Therefore, high quantities of free-form molds are needed to complete free-form architecture. And this results in construction waste, longer construction periods, and higher construction costs. Therefore, the productivity of FCP was required to improve.

To solve this issue, Lee developed an FCP production machine as shown in Figure 1A [8]. The FCP production machine was made up of the side form control equipment and double-side CNC(Computerized Numeric Control) machine. The double-side CNC machine was made up of the upper CNC machine and lower CNC machine. The FCP production machine produced FCP in the following order. First, the silicon rubber was placed on top of the lower CNC machine and the FCP's design form was entered. The rod moved up and down to create the FCP lower curve according to the design form. Similarly, the rod of the side control equipment and upper CNC machine moved to place the concrete in the configured design form to make various FCP shapes as shown in (B). However, a side mold that can be used in the side form control equipment as shown in (C) was needed to realize this.

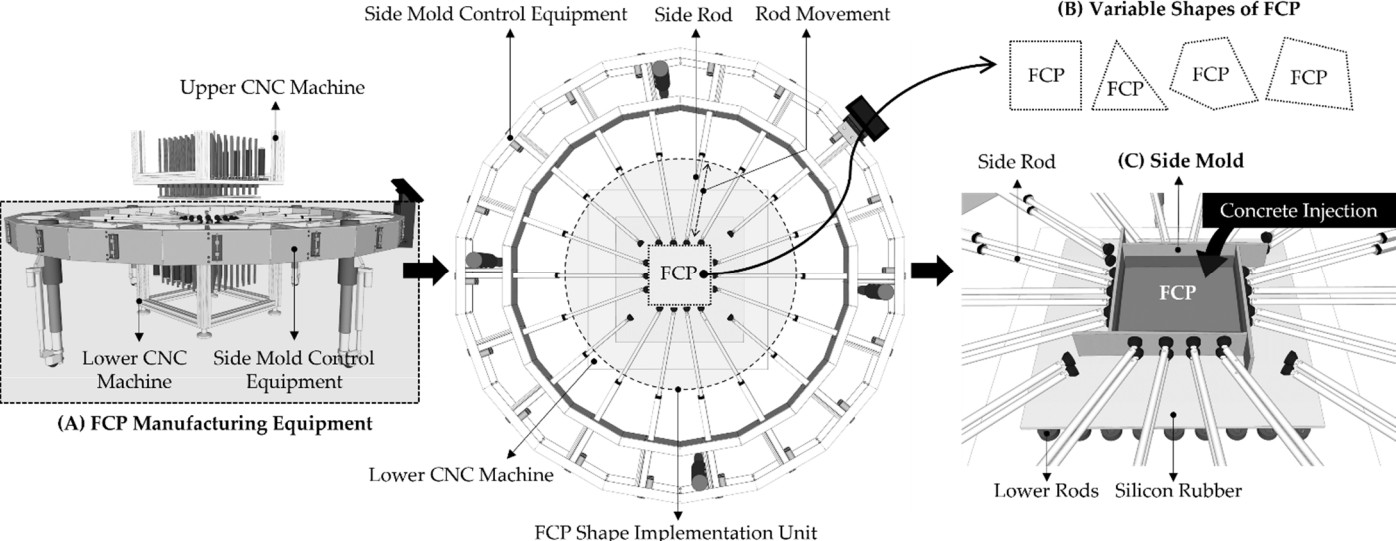

**Figure 1.** FCP Manufacturing Technology ((**A**): FCP Manufacturing Equipment, (**B**): Variable Shapes of FCP, (**C**): Side Mold) [8,9].

FCP requires different curves, lengths and formation, and the number of sides changes depending on the shape. The side mold must be variable and flexible to configure to the form of the FCP. Therefore, this study developed the Variable Side Mold. Prior to its development, the required performance of FCP and past studies were analyzed, and the side mold was developed based on this. The FCP was produced using the developed variable side mold. The margin of error was analyzed by comparing the completed FCP form with the design form. The effectiveness of the variable side mold was verified to develop the side mold for FCP production that can be made with curves.

## 2. Literature Review

FCP refers to a free-form concrete panel that is made into a panel with a shape that can easily produce massive free-form architecture. Configure the envelope of the massive free-form architecture by binding such FCPs. Therefore, the produced free-form panel must be precisely bound. However, if there is an error between the free-form mold and design form, the produced FCP cannot be used. This will result in construction waste and because the free-form mold cannot be used, it will have to be treated, and additional costs will occur. There is also the problem of delayed construction time and increased construction costs due to having to produce a new free-form mold. Therefore, in the case of the FCP production sector, studies using various materials based on CNC are being conducted such as Expandable Polystyrene (EP) [10,11], Phase Change Material (PCM) [12–14], wood [15], fiber [16,17], ice [18], etc.

Oesterle developed a wax mold that can be recycled as it uses wax. The production process is as follows: Wax was melted into liquid shape and was placed in a pre-made mold.

Here, the mold for one side was made with wax and the other side was produced using the same method to complete the wax mold [19]. Concrete was placed in such a mold to create the free-form concrete panel. However, there was no solution against the hardening time of wax, intensity, cracking and crystallization, and the panel's form precision was low, thus making it basic research that will be difficult to commercialize.

As a technology for producing free-form concrete panels, Angelos Savvides developed Free-Form Formwork System Technology (3FST). The 3FST installed 26 rods set apart at a constant interval on both sides, and this system was used to produce 1 m × 1 m free-form concrete panels [20]. The system used a round rod to configure a flexible free-form surface, but unnecessary marks are left behind on the produced free-form concrete panels, thus causing an error on the form. With such technologies, precise form configuration was difficult when producing free-form concrete panels, and as a result, finishing work was additionally required.

Lee (2015) developed a technology that produced FCPs with a CNC Machine. The CNC Machine was combined with silicone rubber made up of several Numerical Control Rods (NCR). Silicon rubber has strong durability and it is a material for which forming is easy. The NCR combined with silicone rubber was moved up and down according to the mold design form to freely configure the bottom side of the FCP [13]. Therefore, production and installation technologies of free-form concrete panels using the CNC machine were developed.

Several studies were conducted with the goal of configuring the bottom side of the free-form mold, but there are few studies related to the side. Yun (2021) developed a side mold control equipment among machines that automatically produce FCPs. This machine fixed the side mold design according to the form of the side of the panel to be produced using a rod. The side mold form changes depending on the panel form, and the angle also changes accordingly. The side mold control equipment was designed circularly so that the rod can be properly fixed on any location depending on the panel shape. The side mold was strongly fixed without the rods interfering with each other and produces various forms automatically [21,22]. However, because the form of the side mold was in a straight line, experimentation on free-from side molds could not be carried out. FCPs have different shapes, sizes, and curves, so a side mold that can configure various shapes is needed. Due to the limitations of existing technologies, free-form molds cannot be reused, and this results in an increased need for manpower, construction delays, and more construction costs. Therefore, technologies related to FCP side molds were lacking.

## 3. Development of Variable Side Mold

### 3.1. Requirement Analysis of Variable Side Mold

Of the FCP production equipment described above, a variable side mold that can be used in side mold control equipment was developed in this study. This required the resolution of limitations of existing free-form molds, and it must be able to precisely configure the side form of FCPs. Therefore, this study aims at analyzing the requirements of variable side molds for development.

First, variable side molds must be variable as the FCP sides have various curves. As existing free-form molds are produced in fixed shapes according to the form, they cannot be reused. FCPs have different forms depending on their used location from basic triangles to polygons, and it must therefore be possible to transform variably according to the form of the FCP. Second, it must be possible to change length according to the side length. The panel size is diverse, and as a result it must be possible to flexibly adjust length for reuse. Third, the thickness of the produced FCP must have even thickness throughout the entire side section of the panel produced. If panels are not produced at an even thickness, errors can occur on the binding parts of the side, thereby not adhering closely and thus making it impossible to use. Fourth, it must have intensity that can sufficiently resist the side pressure of concrete. Strong side pressure occurs on the side due to the free-form. If the intensity was unable to sufficiently show resistance, it can cause the mold form to collapse, resulting in errors in the design form. It is thus necessary to select material with intensity that can

resist side pressure. Lastly, variable side molds must be able to combine easily with the aforementioned FCP side mold control equipment. The mold produced in this study was used in FCP production equipment, and it should therefore be possible to combine with FCP side mold control equipment. Accordingly, it must be produced using materials that can combine with rods in side mold control equipment. These five requirements can be categorized into two types.

Figure 2A shows the shape of an FCP triangle, which is a basic production, but also having different forms and curves. In addition, as the lengths of each side are different, it must be possible to change the length of the side mold. Figure 2B represents even thickness of the side mold. This can make the thickness of the side form of the produced FCP even. Therefore, the first to third requirements as shown below are on side free-form molds for producing FCPs.

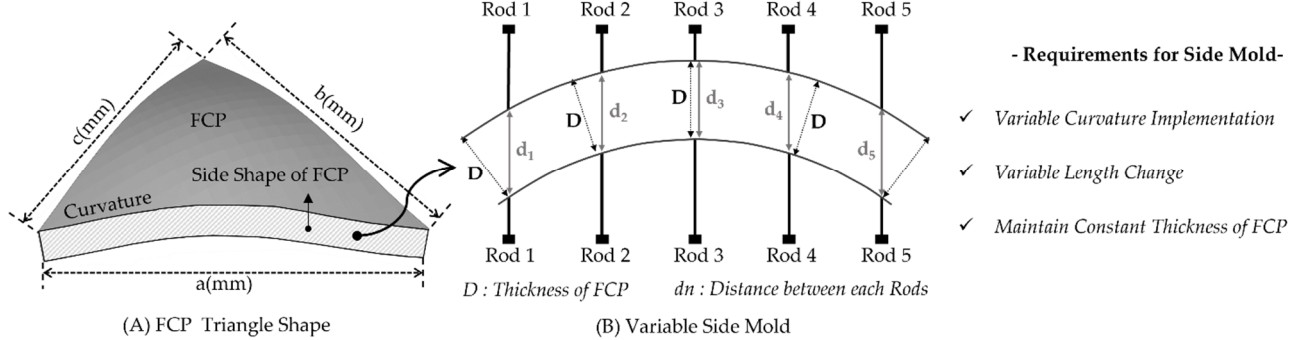

**Figure 2.** Requirements of variable side mold for producing FCPs. ((**A**): FCP Triangle Shape, (**B**): Variable Side Mold).

Figure 3 shows the process of the side mold control equipment producing a FCP. The side rod was combined with the side mold to configure the design form. Thus, when concrete was placed, side pressure as shown in (A) is generated on the side mold. Side molds must be produced with materials that can resist such side pressure to prevent errors with design forms. In addition, it must be a material that allows close adhesion with side rods. As a result, the two demands on side mold materials are as follows.

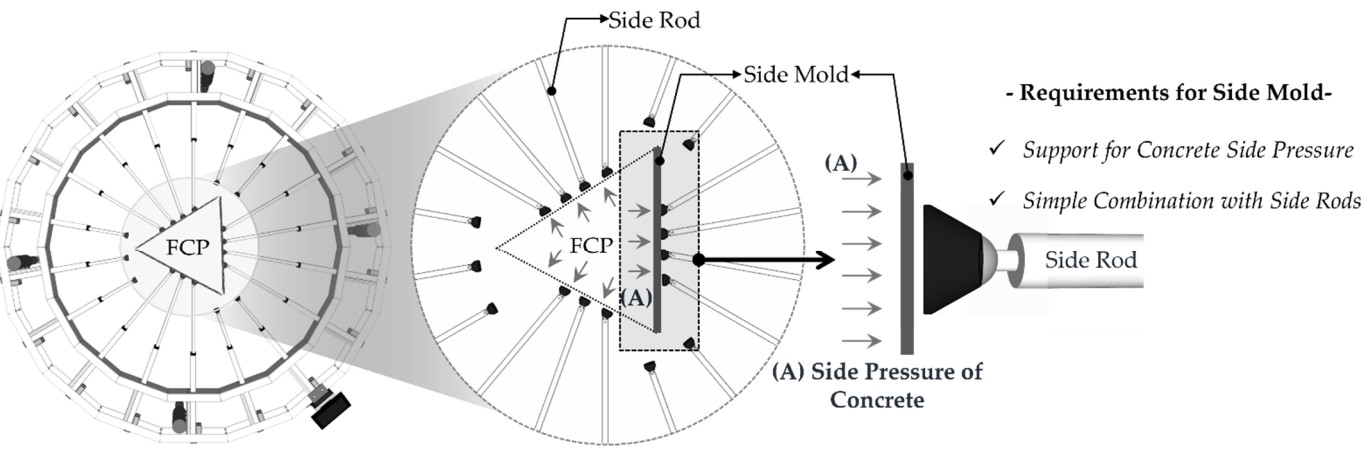

**Figure 3.** Requirements for variable side mold materials.

### 3.2. Equipment Design

It was designed as shown in Figure 4 by satisfying the variable side mold requirements analyzed through the aforementioned Section 3.1. The variable side mold has a structure in which the Lower plate(B) is combined between two Upper plates(A) through

the Join Ring (C). The (A) and (B) are made up of a front plate and rear plate and are bonded with a bolt at the connecting part (D). The Join Ring has a straight elliptical structure and as movement between the (A) and (B) was possible, variable transformation is easy. It has a structure that allows additional connections for (A) and (B) depending on the length of the FCP side form, and this allowed length adjustment of the side mold.

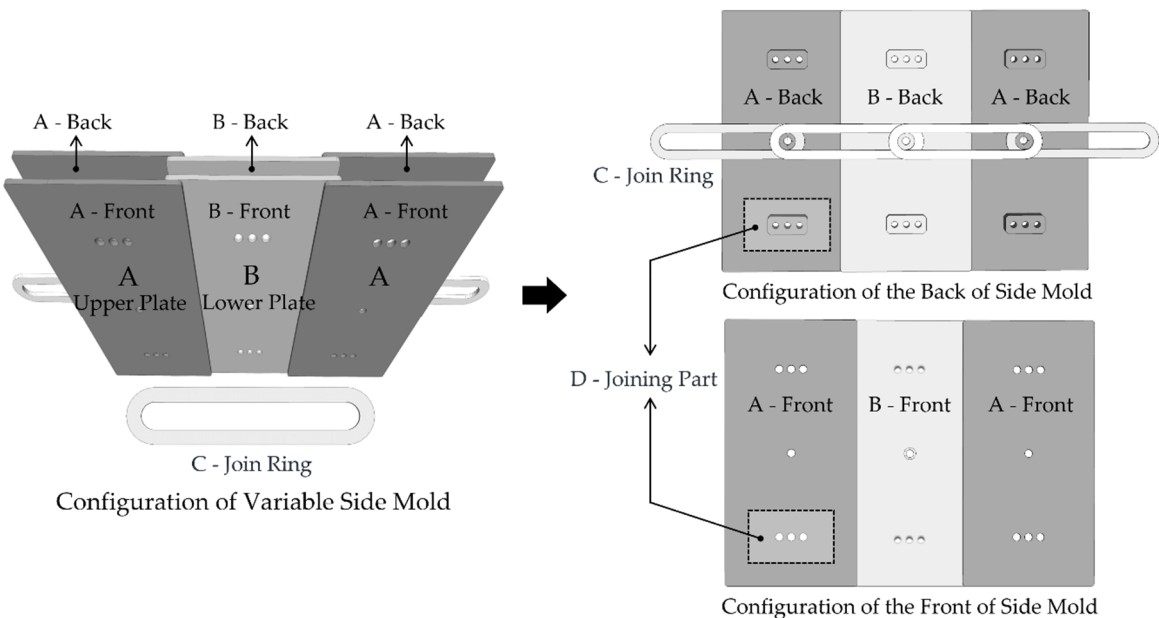

**Figure 4.** Variable side mold length adjustment method. (A: Upper Plate, B: Lower Plate, C: Join Ring, D: Joining Part).

The variable side mold has a structure that connects straight lines and as shown in Figure 5a is a variable side mold transformed according to the form of the FCP side. This makes it possible to create even thickness of 100 mm in any form. Figure 5b shows the side mold control equipment and variable side mold being combined. The variable side mold was produced with steel plates, and therefore has sufficient intensity to resist side pressure that occurs on the side. A magnet was installed on the end of the side rod of the side mold control equipment as shown in (A) to bond with the variable side mold's steel plate material using magnetic force. It can be fixed easily with this detachable connection method.

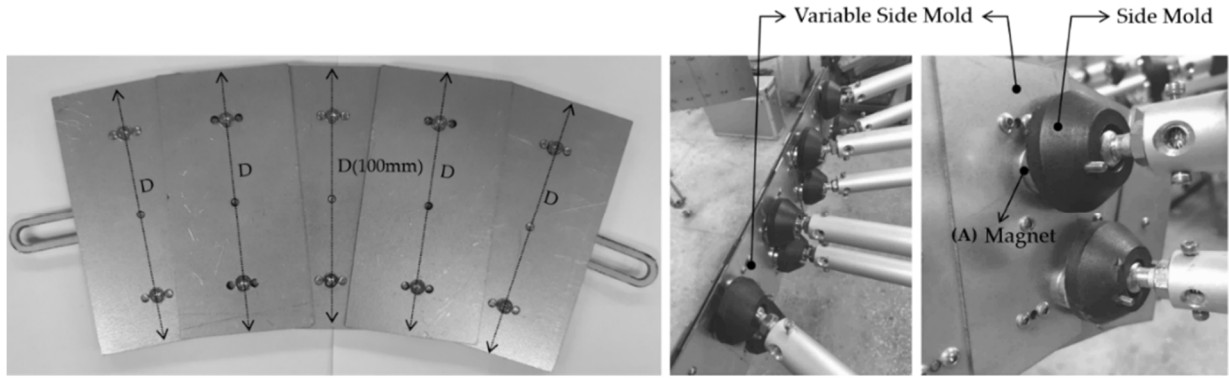

**Figure 5.** Variable side mold and side mold control equipment bonding method.

## 4. FCP Manufacturing Experiment Using Variable Side Mold

This section may be divided by subheadings. It should provide a concise and precise description of the experimental results, their interpretation, as well as the experimental conclusions that can be drawn.

### 4.1. Variable Side Mold Testing Process

The variable side mold was produced satisfying the five aforementioned requirements. This was made up of steel plate materials and required verification on whether or not it has the sufficient intensity that can resist concrete side pressure. Therefore, in order to verify side pressure resistance of the variable side mold, this study aimed at analyzing the error of the side form by producing FCPs.

A rectangular 400 mm × 400 mm × 100 mm FCP was produced as shown in Figure 6 for testing. The variable side mold was installed according to the design form on top of the lower CNC. The side angle of the produced FCP was 79.3° and the max height was 20 mm. This was a setting for measuring resistance of the placed concrete against side pressure for the variable side mold. The objective of the test is to check the error in the produced FCP side form to verify the side pressure resistance of the variable side mold.

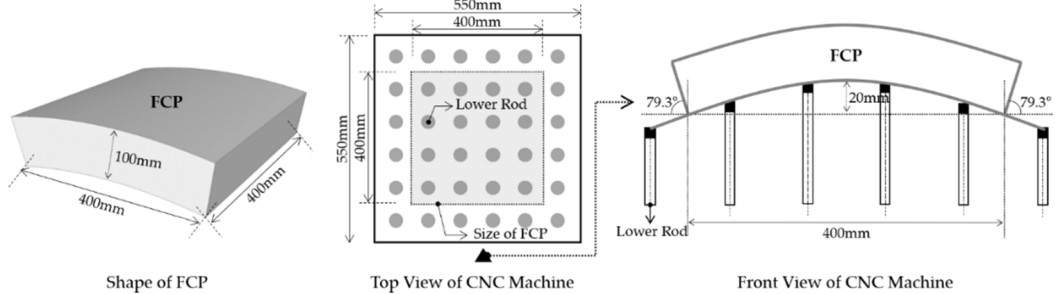

**Figure 6.** Variable side mold test settings.

The variable side mold test process is as shown in Figure 7. The lower CNC moved the rod up and down to transform the silicon plate according to the design form. After installing the variable side mold at the curvature, according to the design form on top of the transformed silicon plate, it was fixed with a hinge. The actual variable side mold was fixed to the rod of the side mold control equipment with a hinge. However, in this study, tests were conducted without side mold control equipment to check the side pressure resistance of the variable side mold. The hinge is a basic part used together with side mold control equipment with the goal of maintaining the form of the mold. This lacks enough intensity to maintain the form of the mold without side mold control equipment. Thus, in order to prevent the collapse of the variable side mode during testing, Expanded Poly Styrene (EPS) was bonded to supplement the lacking fixing device. Afterwards, concrete was placed and hardened, and then the mold was disassembled to produce the FCP.

### 4.2. FCP Form Error Analysis

The variable side mold has a structure which connects to the steel plate, and this realizes even thickness in all areas to transform according to the design form to produce the FCP. The FCP produced was checked to see whether the FCP with the variable side mold was the same as the design form. Thus, the design form and FCP form were compared through 3D scanning and quality inspections to verify whether errors occurred.

When errors occurred on the FCP side, the construction errors occurred when assembling them. However, as it was impossible to produce all panels according to design form without errors in massive free-form architecture projects, some margins of error have to be accepted. In this study, there were no criteria for allowed margin of errors for FCP production; therefore, quality inspections were performed on panel side errors based on wall quality standards. The error that occurred by comparing the produced and FCP with the design form through 3D

scanning is as shown in (Figure 8). It was divided into a total of four zones, namely B (b, b1), C (c, c1), and D (d, d1) with A as the center for the produced panel side. Zone D is where the EPS form is fixed, so it is the same as the design form. Results of tests showed that there are minor errors, but it was close to 0 and there were no issues with precision of the variable side mold. Zone A, B, and C can measure intensity only for the variable side mold. Therefore, 40 data per zone were extracted to find the error measurement.

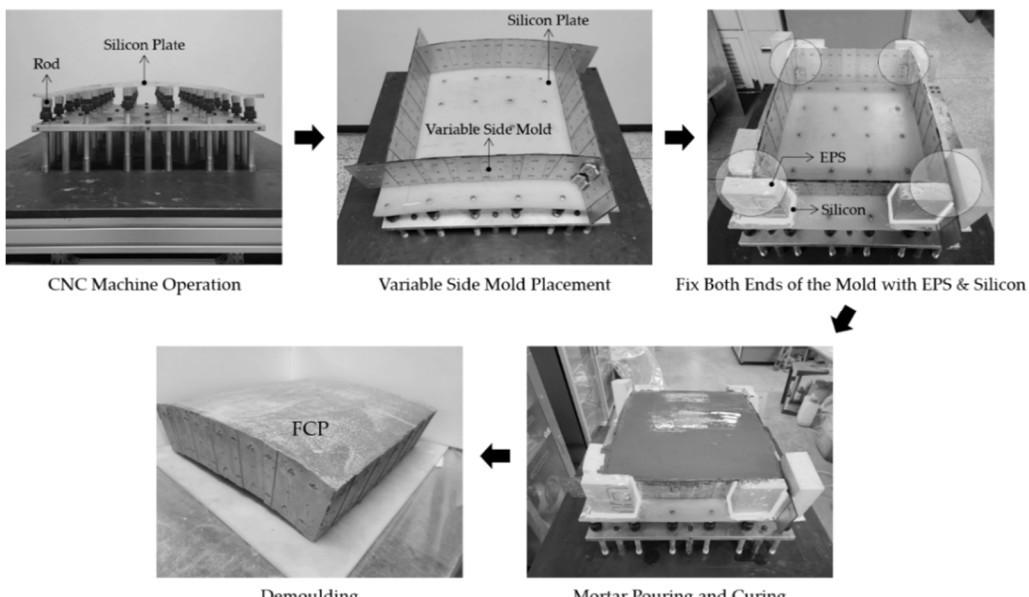

**Figure 7.** FCP production test process using variable side mold.

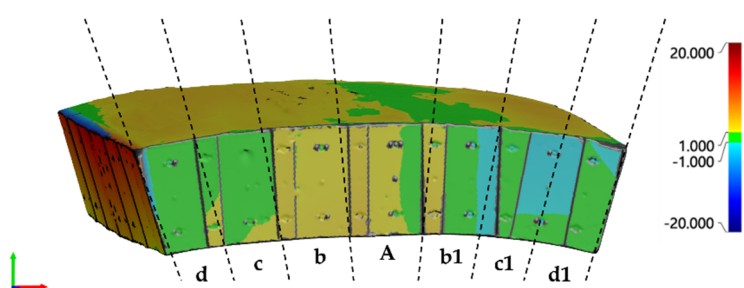

**Figure 8.** Side error of FCP based on design form. (Unit: mm).

This variable side mold was made up of several pieces for free manifestation of forms and it required connection for each piece. Such structure effectively configures free forms, but there was tolerance in the connecting part which can therefore result in errors. In order to check such errors, the error was measured after producing the free-form panel with the variable side mold. The analysis compared errors that occurred on each end and in the center of the mold. The ends of both sides have no differences with design forms even if concrete side pressure occurs due to fixing devices. However, the central part of the mold has no fixing device, and as a result the only element that can resist the side pressure of concrete is the intensity of the mold. Accordingly, when there was a difference in errors between both ends and the center of the mold, this means that the steel plate materials used in the variable side mold had insufficient intensity for resisting the side pressure of concrete.

Table 1 shows descriptive statistics for error measurement values of the four zones on the sides of the produced panels. The average error of the four zones was "0.4309" mm, "0.4998" mm, "0.5905" mm, and "0.7079" mm; all zones exhibited similar std. deviation of about "0.39". The mean difference of the center and both ends was "0.276" mm, but

the variance of the0 measurements was large and there is also a high average difference of maximum and minimum values. Thus, in order to verify the statistical significance of average differences, the *t*-test was performed. The features of the errors generated were checked through this.

**Table 1.** Descriptive statistics of produced FCP errors.

| Area | N | Mean | Std. Deviation | Std. Error | 95% Confidence Interval for Mean | | Minimum | Maximum |
|---|---|---|---|---|---|---|---|---|
| | | | | | Lower Bound | Upper Bound | | |
| A | 40 | 0.4309 | 0.4036 | 0.0638 | 0.3019 | 0.5600 | 0.01 | 1.34 |
| B | 40 | 0.4998 | 0.3927 | 0.0621 | 0.3742 | 0.6254 | 0.01 | 1.34 |
| C | 40 | 0.5905 | 0.3253 | 0.0514 | 0.4864 | 0.6945 | 0.06 | 1.51 |
| D | 40 | 0.7079 | 0.3567 | 0.0564 | 0.5938 | 0.8219 | 0.27 | 1.60 |
| Total | 160 | 0.5573 | 0.3818 | 0.0302 | 0.4976 | 0.6169 | 0.01 | 1.60 |

The null hypothesis of the *t*-test is presumed as follows:

**Hypothesis H1.** *There is no difference in the error of the two groups (central, ends).*

**Hypothesis H2.** *There is a difference in the error of the two groups (central, ends).*

Results of the *t*-test show that *t*-value is "−3.252" and *p*-value is "0.002" as shown in Table 2; therefore, H1 can be dismissed in the two tailed test. In other words, it can be judged that there is a difference in error per part at a 95% confidence level. This represents that there were errors in the ends and center of the mold.

**Table 2.** *t*-Test results.

| *t* | df | Sig (2-Tailed) | Mean Difference | Std. Error Difference | 95% Confidence Interval of the Difference | |
|---|---|---|---|---|---|---|
| | | | | | Lower | Upper |
| −3.252 | 78 | 0.002 | −0.27695 | 0.08516 | −0.44648 | −0.10742 |

As supports were installed on both ends of the panel, resistance to side pressure was possible, and error was close to 0. However, errors occurred on the two ends. This is because there was a difference in design form in the course of installing the mold manually. Accordingly, when comparing the error between zone D, which was the two ends, and zone A, which was the center, it was found that the error became more proportional when distance from the support increased, rather than due to lacking intensity of the mold. This indicates that errors can increase due to space around the support. therefore, by adjusting the distance from supports according to the range of allowed errors when producing panels there will be no issues for configuring the form.

The deformable side formwork has a structure in which an inner plate is coupled between two outer plates through a coupling ring. This structure helps to freely manufacture the shape of the panel. In addition, it was confirmed that errors were within the allowable range when manufacturing the formwork with this method. However, a prerequisite is sufficient rigidity of the formwork material. Additionally, distortion of the mold material can be a source of error. The use of this mold in practice requires attention to stiffness and warping.

## 5. Conclusions

In this study, a variable side mold was developed allowing for the configuration of side forms of FCPs. The variable side mold satisfied the five requirements with the goal of utilization in existing FCP production equipment. The variable side mold can be variably transformed instead of using a fixed form to make various FCP side forms. The mold can

be disassembled and assembled according to the side length to flexibly change the length according to the panel size. Thickness must be even in all sectors of the side to prevent errors from occurring in the connecting parts of the panel. It was produced in a structure of combined straight lines that can change the height of the variable side mold to configure a constant 100mm thickness for all sectors. It was made up of steel plate materials of the variable side mold, and among the aforementioned FCP production equipment, it was bound to the rod of the side mold control equipment with magnetic force. Furthermore, when producing actual panels, side pressure is generated in the mold in the concrete placing process. The question of whether the variable side mold has sufficient intensity for resistance was verified through the FCP production test.

For the FCP production test, the panel side was divided into four zones and the 40 values of each zone were summed up to find descriptive statistics for error measurement. All zones exhibited similar std. deviation at approximately 0.39. The mean difference between the central part and each end was 0.276 mm and the *t*-test was performed to check statistical significance. In the results, errors occurred at the two ends and central part at a 95% confidence level. This error resulted not because of lacking mold intensity, but due to the absence of a support; therefore, it was judged to be a result of the spacing of supports in the variable side mold. Accordingly, as the variable side mold was combined with the rod of the side mold control equipment, if the rod bonds on the location where side pressure is applied, the margin of error can be reduced. As the variable side mold resists concrete side pressure, it was proven to have sufficient intensity. The results of this study are expected to be used as a basis for research on configuring the side form of free-form panels, and it will help improve the reuse and productivity of molds. In the future, research for minimizing errors that occur when producing FCPs using this equipment, while combining with side mold control equipment, should be conducted.

**Author Contributions:** Conceptualization, J.Y. (Jongyoung Youn) and J.Y. (Jiyeong Yun); methodology, J.Y. (Jongyoung Youn); validation, J.Y. (Jongyoung Youn), J.Y. (Jiyeong Yun), and D.L.; formal analysis, J.Y. (Jongyoung Youn); investigation, J.K.; writing—original draft preparation, J.Y. (Jongyoung Youn); writing—review and editing, D.L.; project administration, J.Y. (Jongyoung Youn) All authors have read and agreed to the published version of the manuscript.

**Funding:** This work was supported by the National Research Foundation of Korea (NRF) grant funded by the Korea government (MSIT) (No. 2020R1C1C1012600).

**Data Availability Statement:** Not applicable.

**Conflicts of Interest:** The authors declare no conflict of interest.

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
