# Peer review of "Development of Variable Side Mold for Free-Form Concrete Panel Production"

_buildings, doi:10.3390/buildings12060728_

Round 1

Reviewer 1 Report

This paper presents a concise yet interesting report on the development of equipment to resolve the low precision of shape in free-form concrete panels (FCP) production. In general, the work follows a logical sequence. Sections 1 and 2 present sufficient contextualization and theoretical background. The methodology is well described, and the findings are clearly presented. Some minor comments should be addressed prior to publication:

  • Abbreviations (like FCP) should be avoided in the title.
  • In the Abstract, the authors mention "In particular, as free-form panels are produced manually". I don't think "manually" is the correct term. Perhaps "handicraft" or something that brings the idea of "artisanal".
  • Table 2 should have a label of the terms "t", "df" and "Sig 2-tailed"
  • Units are missing in Figure 8
  • The Conclusion section should be shortened. 

Author Response

The authors would like to first thank the editor who allowed us opportunities to revise and resubmit the paper. We also sincerely appreciate the anonymous reviewer who provided thorough reviews and valuable comments to help us improve the manuscript. We strongly believe that in the revision we have fully addressed all the reviewer’s comments and concerns and carefully revised the manuscript based on the feedback we have received. Please see the followings below responding to reviewer’s comments.

Reviewer comments.

This paper presents a concise yet interesting report on the development of equipment to resolve the low precision of shape in free-form concrete panels (FCP) production. In general, the work follows a logical sequence. Sections 1 and 2 present sufficient contextualization and theoretical background. The methodology is well described, and the findings are clearly presented. Some minor comments should be addressed prior to publication:

Abbreviations (like FCP) should be avoided in the title.

Reply. Abbreviations was deleted in the title.

Original. [Title] Development of Variable Side Mold for FCP Production

Revised. [Title] Development of Variable Side Mold for Free-form Concrete Panel Production

Reviewer comments. In the Abstract, the authors mention "In particular, as free-form panels are produced manually". I don't think "manually" is the correct term. Perhaps "handicraft" or something that brings the idea of "artisanal".

Reply. We corrected that as below.

Original. [In abstract] In particular, as free-form panels are produced manually

Revised. [In abstract] In particular, as free-form panels are produced handicraft.

Reviewer comments. Units are missing in Figure 8

Reply. We added the unit as below.

Added. [Figure 8] (Unit : mm)

Reviewer comments. The Conclusion section should be shortened.

Reply. We revised the conclusion as below.

Original. [5. Conclusion] In this study, a variable side mold was developed allowing the configuration of side forms of FCPs. The variable side mold satisfied the five requirements with the goal of utilizing in the existing FCP production equipment. The variable side mold can be variably transformed instead of using a fixed form to make various FCP side forms. The mold can be disassembled and assembled according to the side length to flexibly change the length according to the panel size. Thickness must be even in all sectors of the side to prevent errors from occurring in the connecting parts of the panel. It was produced in a structure of combining straight lines that can change the height of the variable side mold to configure a constant 100mm thickness for all sectors. It was made up of steel plate materials of the variable side mold, and among the aforementioned FCP production equipment, it was bound to the rod of the side mold control equipment with magnetic force. Furthermore, when producing actual panels, side pressure is generated in the mold in the concrete placing process. The question on whether the Variable Side Mold has sufficient intensity for resistance was verified through the FCP production test.

The FCP production test process used the CNC machine and variable side mold. The variable side mold was installed on the lower CNC set according to the design form and it is fixed with a hinge. This test did not use a side mold control equipment to verify the intensity of the variable side mold. Therefore, as it was difficult to conduct the test with a hinge, additional EPS form was installed on the two ends of the mold. Afterwards, the FCP produced by placing concrete was measured for differences between the design form and produced form through 3D scanning and quality inspections to conduct form error analysis. The analysis process involved analyzing the difference of errors in the central part that resisted only with the mold without a fixing device on the ends fastened with a hinge and EPS form. The panel side was divided in-to four zones and the 40 values of each zone were summed up to find the descriptive statistics for error measurements. All zones exhibited similar std. deviation at ap-proximately 0.39. The mean difference between the central part and each end was 0.276mm and t-test was performed to check the statistical significance. In result, errors occurred at the two ends and central part a 95% confidence level. This error resulted not because of lacking mold intensity, but due to the absence of a support, and there-fore, it was judged to be a result of the space of supports in the variable side mold. Accordingly, as the variable side mold was combined with the rod of the side mold control equipment, if the rod bonds on the location where side pressure is applied, the margin of error can be reduced. As the variable side mold resists concrete side pressure, it was proven to have sufficient intensity. The results of this study are expected to be used as basic research for configuring the side form of free-form panels, and it will help improve the reuse and productivity of molds. In the future, research for minimizing error that occurs when producing FCP using this equipment, while combining with the side mold control equipment should be conducted.

Revised. [5. Conclusion] In this study, a variable side mold was developed allowing the configuration of side forms of FCPs. The variable side mold satisfied the five requirements with the goal of utilizing in the existing FCP production equipment. The variable side mold can be variably transformed instead of using a fixed form to make various FCP side forms. The mold can be disassembled and assembled according to the side length to flexibly change the length according to the panel size. Thickness must be even in all sectors of the side to prevent errors from occurring in the connecting parts of the panel. It was produced in a structure of combining straight lines that can change the height of the variable side mold to configure a constant 100mm thickness for all sectors. It was made up of steel plate materials of the variable side mold, and among the aforementioned FCP production equipment, it was bound to the rod of the side mold control equipment with magnetic force. Furthermore, when producing actual panels, side pressure is generated in the mold in the concrete placing process. The question on whether the Variable Side Mold has sufficient intensity for resistance was verified through the FCP production test.

For the FCP production test the panel side was divided into four zones and the 40 values of each zone were summed up to find the descriptive statistics for error measurements. All zones exhibited similar std. deviation at approximately 0.39. The mean difference between the central part and each end was 0.276mm and t-test was performed to check the statistical significance. In result, errors occurred at the two ends and central part a 95% confidence level. This error resulted not because of lacking mold intensity, but due to the absence of a support, and therefore, it was judged to be a result of the space of supports in the variable side mold. Accordingly, as the variable side mold was combined with the rod of the side mold control equipment, if the rod bonds on the location where side pressure is applied, the margin of error can be reduced. As the variable side mold resists concrete side pressure, it was proven to have sufficient intensity. The results of this study are expected to be used as basic research for configuring the side form of free-form panels, and it will help improve the reuse and productivity of molds. In the future, research for minimizing error that occurs when producing FCP using this equipment, while combining with the side mold control equipment should be conducted.

Reviewer 2 Report

The conducted work “Development of Variable Side Mold for FCP Production” is good. However, following comments should be addressed to further improve paper:

  1. It is better not to use abbreviation in title.
  2. Add more recent relevant literature review from 2021 and 2022 in introduction and literature review sections. Also, explicitly mention the novelty and research significance of current work in last paragraph of introduction section.
  3. Avoid long sentences throughout the manuscript, e.g. lines 41-44, etc.
  4. Avoid few sentences short paragraphs throughout the manuscript, e.g. lines 109-112, etc.
  5. Outcome should be further discussed in detail. More scientific reasoning emphasis is required while elaborating outcome.
  6. There should be a separate section (before conclusions section) explaining the implementation of this research in real field for practicing professionals.
  7. Conclusions are long. These should be made brief and to the point. In addition, closing remarks should be added at the end of conclusion section keeping in mind all conclusive bullet points.
  8. English Language should be improved throughout the manuscript.

Author Response

The authors would like to first thank the editor who allowed us opportunities to revise and resubmit the paper. We also sincerely appreciate the anonymous reviewer who provided thorough reviews and valuable comments to help us improve the manuscript. We strongly believe that in the revision we have fully addressed all the reviewer’s comments and concerns and carefully revised the manuscript based on the feedback we have received. Please see the followings below responding to reviewer’s comments.

Reviewer comments.

The conducted work “Development of Variable Side Mold for FCP Production” is good. However, following comments should be addressed to further improve paper:

It is better not to use abbreviation in title.

Reply. Abbreviations was deleted in the title.

Original. [Title] Development of Variable Side Mold for FCP Production

Revised. [Title] Development of Variable Side Mold for Free-form Concrete Panel Production

Reviewer comments. Add more recent relevant literature review from 2021 and 2022 in introduction and literature review sections. Also, explicitly mention the novelty and research significance of current work in last paragraph of introduction section.

Reply. We added more relevant literature which was published in 2021.

Added. [2. Literature review, Line 108] The side mold was strongly fixed without the rods interfering with each other and produces various forms automatically [21,22].

Added. [References, Line 373] 22. Jeong, K. Development of Two-Sided CNC and Side Mould Control Equipment for Automatic Manufacture of Free-form Concrete Panel. Master's thesis, Hanbat National University, Daejeon, Korea, 2021.

Reviewer comments. Avoid long sentences throughout the manuscript, e.g. lines 41-44, etc.

Reply. We revised the sentences as below

Original. [Line 41] Therefore, high quantities of free-form mold are needed to complete free-form architecture, and this results in construction wastes, longer construction periods, and higher construction costs, thus requires improving the productivity of FCP.

Revised. [line 41] Therefore, high quantities of free-form mold are needed to complete free-form architecture. And this results in construction wastes, longer construction periods, and higher construction costs. Therefore, the productivity of FCP was required to improve.

Reviewer comments. Avoid few sentences short paragraphs throughout the manuscript, e.g. lines 109-112, etc.

Reply. We revised the paragraphs.

Revised. [line 100-113] Several studies were conducted with the goal of configuring the bottom side of the free-form mold, but there are few studies related to the side. Yun (2021) developed a side mold control equipment among machines that automatically produce FCPs. This machine fixed the side mold designed according to the form of the side of the panel to be produced using a rod. The side mold form changes depending on the panel form, and the angle also changes accordingly. The side mold control equipment was de-signed circularly so that the rod can be properly fixed on any location depending on the panel shape. The side mold was strongly fixed without the rods interfering with each other and produces various forms automatically [21,22]. But because the form of the side mold was in a straight line, experimentation on side mold of the free-form could not be carried out. FCPs have different shapes, sizes and curves, so a side mold that can configure various shapes is needed. Due to the limitations of existing tech-nologies, free-form molds cannot be reused, and this results in increased need for manpower, construction delays, and more construction costs. Therefore, technologies on FCP side molds were lacking.

Reviewer comments. There should be a separate section (before conclusions section) explaining the implementation of this research in real field for practicing professionals.

Reply. We explained the implementation of this research in real field for practicing professionals as below.

Added. [4.2 FCP Form Error Analysis, Line 269] The deformable side formwork has a structure in which an inner plate is coupled between two outer plates through a coupling ring. This structure helps to freely manufacture the shape of the panel. In addition, it was confirmed that the error was within the allowable range when manufacturing the formwork with this method. However, a prerequisite is sufficient rigidity of the formwork material. Also, distortion of the mold material can be a source of error. The use of this mold in practice requires attention to stiffness and warpage.

Reviewer comments. Conclusions are long. These should be made brief and to the point. In addition, closing remarks should be added at the end of conclusion section keeping in mind all conclusive bullet points.

Reply. We revised the conclusion as below.

Original. [5. Conclusion] In this study, a variable side mold was developed allowing the configuration of side forms of FCPs. The variable side mold satisfied the five requirements with the goal of utilizing in the existing FCP production equipment. The variable side mold can be variably transformed instead of using a fixed form to make various FCP side forms. The mold can be disassembled and assembled according to the side length to flexibly change the length according to the panel size. Thickness must be even in all sectors of the side to prevent errors from occurring in the connecting parts of the panel. It was produced in a structure of combining straight lines that can change the height of the variable side mold to configure a constant 100mm thickness for all sectors. It was made up of steel plate materials of the variable side mold, and among the aforementioned FCP production equipment, it was bound to the rod of the side mold control equipment with magnetic force. Furthermore, when producing actual panels, side pressure is generated in the mold in the concrete placing process. The question on whether the Variable Side Mold has sufficient intensity for resistance was verified through the FCP production test.

The FCP production test process used the CNC machine and variable side mold. The variable side mold was installed on the lower CNC set according to the design form and it is fixed with a hinge. This test did not use a side mold control equipment to verify the intensity of the variable side mold. Therefore, as it was difficult to conduct the test with a hinge, additional EPS form was installed on the two ends of the mold. Afterwards, the FCP produced by placing concrete was measured for differences between the design form and produced form through 3D scanning and quality inspections to conduct form error analysis. The analysis process involved analyzing the difference of errors in the central part that resisted only with the mold without a fixing device on the ends fastened with a hinge and EPS form. The panel side was divided in-to four zones and the 40 values of each zone were summed up to find the descriptive statistics for error measurements. All zones exhibited similar std. deviation at ap-proximately 0.39. The mean difference between the central part and each end was 0.276mm and t-test was performed to check the statistical significance. In result, errors occurred at the two ends and central part a 95% confidence level. This error resulted not because of lacking mold intensity, but due to the absence of a support, and there-fore, it was judged to be a result of the space of supports in the variable side mold. Accordingly, as the variable side mold was combined with the rod of the side mold control equipment, if the rod bonds on the location where side pressure is applied, the margin of error can be reduced. As the variable side mold resists concrete side pressure, it was proven to have sufficient intensity. The results of this study are expected to be used as basic research for configuring the side form of free-form panels, and it will help improve the reuse and productivity of molds. In the future, research for minimizing error that occurs when producing FCP using this equipment, while combining with the side mold control equipment should be conducted.

Revised. [5. Conclusion] In this study, a variable side mold was developed allowing the configuration of side forms of FCPs. The variable side mold satisfied the five requirements with the goal of utilizing in the existing FCP production equipment. The variable side mold can be variably transformed instead of using a fixed form to make various FCP side forms. The mold can be disassembled and assembled according to the side length to flexibly change the length according to the panel size. Thickness must be even in all sectors of the side to prevent errors from occurring in the connecting parts of the panel. It was produced in a structure of combining straight lines that can change the height of the variable side mold to configure a constant 100mm thickness for all sectors. It was made up of steel plate materials of the variable side mold, and among the aforementioned FCP production equipment, it was bound to the rod of the side mold control equipment with magnetic force. Furthermore, when producing actual panels, side pressure is generated in the mold in the concrete placing process. The question on whether the Variable Side Mold has sufficient intensity for resistance was verified through the FCP production test.

For the FCP production test the panel side was divided into four zones and the 40 values of each zone were summed up to find the descriptive statistics for error measurements. All zones exhibited similar std. deviation at approximately 0.39. The mean difference between the central part and each end was 0.276mm and t-test was performed to check the statistical significance. In result, errors occurred at the two ends and central part a 95% confidence level. This error resulted not because of lacking mold intensity, but due to the absence of a support, and therefore, it was judged to be a result of the space of supports in the variable side mold. Accordingly, as the variable side mold was combined with the rod of the side mold control equipment, if the rod bonds on the location where side pressure is applied, the margin of error can be reduced. As the variable side mold resists concrete side pressure, it was proven to have sufficient intensity. The results of this study are expected to be used as basic research for configuring the side form of free-form panels, and it will help improve the reuse and productivity of molds. In the future, research for minimizing error that occurs when producing FCP using this equipment, while combining with the side mold control equipment should be conducted.

Reviewer comments. English Language should be improved throughout the manuscript.

Reply. We improved English language throughout the manuscript.